# IMPROVED TRAINING TECHNIQUE FOR LATENT CONSISTENCY MODELS

**Quan Dao**[*][†]
Rutgers University
quan.dao@rutgers.edu

**Khanh Doan**[*]
Qualcomm Vietnam Company Limited
khanhdn@qti.qualcomm.com

**Di Liu**
Rutgers University
di.liu@rutgers.edu

**Trung Le**
Monash University
trunglm@monash.edu

**Dimitris Metaxas**
Rutgers University
dnm@cs.rutgers.edu

## ABSTRACT

Consistency models are a new family of generative models capable of producing high-quality samples in either a single step or multiple steps. Recently, consistency models have demonstrated impressive performance, achieving results on par with diffusion models in the pixel space. However, the success of scaling consistency training to large-scale datasets, particularly for text-to-image and video generation tasks, is determined by performance in the latent space. In this work, we analyze the statistical differences between pixel and latent spaces, discovering that latent data often contains highly impulsive outliers, which significantly degrade the performance of iCT in the latent space. To address this, we replace Pseudo-Huber losses with Cauchy losses, effectively mitigating the impact of outliers. Additionally, we introduce a diffusion loss at early timesteps and employ optimal transport (OT) coupling to further enhance performance. Lastly, we introduce the adaptive scaling-$c$ scheduler to manage the robust training process and adopt Non-scaling LayerNorm in the architecture to better capture the statistics of the features and reduce outlier impact. With these strategies, we successfully train latent consistency models capable of high-quality sampling with one or two steps, significantly narrowing the performance gap between latent consistency and diffusion models. The implementation is released here: https://github.com/quandao10/sLCT/

## 1 INTRODUCTION

In recent years, generative models have gained significant prominence, with models like ChatGPT excelling in language generation and Stable Diffusion (Rombach et al., 2021). In computer vision, the diffusion model (Song et al., 2020; Song & Ermon, 2019; Ho et al., 2020; Sohl-Dickstein et al., 2015) has quickly popularized and dominated the Adversarial Generative Model (GAN) (Goodfellow et al., 2014). It is capable of generating high-quality diverse images that beat SoTA GAN models (Dhariwal & Nichol, 2021). Additionally, diffusion models are easier to train, as they avoid the common pitfalls of training instability and the need for meticulous hyperparameter tuning associated with GANs. The application of diffusion spans the entire computer vision field, including text-to-image generation (Rombach et al., 2021; Gu et al., 2022), image editing (Meng et al., 2021; Wu & la Torre, 2023; Huberman-Spiegelglas et al., 2024; Han et al., 2024; He et al., 2024), text-to-3D generation (Poole et al., 2022; Wang et al., 2024), personalization (Ruiz et al., 2022; Van Le et al., 2023; Kumari et al., 2023) and control generation (Zhang et al., 2023b; Brooks et al., 2023; Zhangli et al., 2024). Despite their powerful capabilities, they require thousands of function evaluations for sampling, which is computationally expensive and hinders their application in the real world. Numerous efforts have been made to address this sampling challenge, either by proposing new training frameworks (Xiao et al., 2021; Rombach et al., 2021) or through distillation techniques

---

[*]Equal contributions
[†]Project Lead & Corresponding Author

(Meng et al., 2023; Yin et al., 2024; Sauer et al., 2023; Dao et al., 2024a). However, methods like (Xiao et al., 2021) suffer from low recall due to the inherent challenges of GAN training, while (Rombach et al., 2021) still requires multi-step sampling. Distillation-based approaches, on the other hand, rely heavily on pretrained diffusion models and demand additional training.

Recently, (Song et al., 2023) introduced a new family of generative models called the consistency model. Compared to the diffusion model (Song & Ermon, 2019; Song et al., 2020; Ho et al., 2020), the consistency model could both generate high-quality samples in a single step and multi-steps. The consistency model could be obtained by either consistency distillation (CD) or consistency training (CT). In previous work (Song et al., 2023), CD significantly outperforms CT. However, the CD requires additional training budget for using pretrained diffusion, and its generation quality is inherently limited by the pretrained diffusion. Subsequent research (Song & Dhariwal, 2023) improves the consistency training procedure, resulting in performance that not only surpasses consistency distillation but also approaches SoTA performance of diffusion models. Additionally, several works (Kim et al., 2023; Geng et al., 2024) have further enhanced the efficiency and performance of CT, achieving significant results. However, all of these efforts have focused exclusively on pixel space, where data is perfectly bounded. In contrast, most large-scale applications of diffusion models, such as text-to-image or video generation, operate in latent space (Rombach et al., 2021; Gu et al., 2022), as training on pixel space for large-scale datasets is impractical. Therefore, to scale consistency models for large datasets, the consistency must perform effectively in latent space. This work addresses the key question: How well can consistency models perform in latent space? To explore this, we first directly applied the SoTA pixel consistency training method, iCT (Song & Dhariwal, 2023), to latent space. The preliminary results were extremely poor, as illustrated in fig. 5, motivating a deeper investigation into the underlying causes of this suboptimal performance. We aim to improve CT in latent space, narrowing the gap between the performance of latent consistency and diffusion.

We first conducted a statistical analysis of both latent and pixel spaces. Our analysis revealed that the latent space contains impulsive outliers, which, while accounting for a very small proportion, exhibit extremely high values akin to salt-and-pepper noise. We also drew a parallel between Deep Q-Networks (DQN) and the Consistency Model, as both employ temporal difference (TD) loss. This could lead to training instability compared to the Kullback-Leibler (KL) loss used in diffusion models. Even in bounded pixel space, the TD loss still contains impulsive outliers, which (Song & Dhariwal, 2023) addressed by proposing the use of Pseudo-Huber loss to reduce training instability. As shown in fig. 1, the latent input contains extremely high impulsive outliers, leading to very large TD values. Consequently, the Pseudo-Huber loss fails to sufficiently mitigate these outliers, resulting in poor performance as demonstrated in fig. 5. To overcome this challenge, we adopt Cauchy loss, which heavily penalizes extremely impulsive outliers. Additionally, we introduce diffusion loss at early timesteps along with optimal transport (OT) matching, both of which significantly enhance the model's performance. Finally, we propose an adaptive scaling $c$ schedule to effectively control the robustness of the model, and we incorporate Non-scaling LayerNorm into the architecture. With these techniques, we significantly boost the performance of latent consistency model compared to the baseline iCT framework and bridge the gap between the latent diffusion and consistency training.

## 2 RELATED WORKS

Consistency model (Song et al., 2023; Song & Dhariwal, 2023) proposes a new type of generative model based on PF-ODE, which allows 1, 2 or multi-step sampling. The consistency model could be obtained by either training from scratch using an unbiased score estimator or distilling from a pretrained diffusion model. Several works improve the training of the consistency model. ACT (Kong et al., 2023), CTM (Kim et al., 2023) propose to use additional GAN along with consistency objective. While these methods could improve the performance of consistency training, they require an additional discriminator, which could need to tune the hyperparameters carefully. MCM (Heek et al., 2024) introduces multistep consistency training, which is a combination of TRACT (Berthelot et al., 2023) and CM (Song et al., 2023). MCM increases the sampling budget to 2-8 steps to tradeoff with efficient training and high-quality image generation. ECM (Geng et al., 2024) initializes the consistency model by pretrained diffusion model and fine-tuning it using the consistency training objective. ECM vastly achieves improved training times while maintaining good generation performance. However, ECM requires pretrained diffusion model, which must use the same architecture as the pretrained diffusion architecture. Although these works successfully improve the

performance and efficiency of consistency training, they only investigate consistency training on pixel space. As in the diffusion model, where most applications are now based on latent space, scaling the consistency training (Song et al., 2023; Song & Dhariwal, 2023) to text-to-image or higher resolution generation requires latent space training. Otherwise, with pretrained diffusion model, we could either finetune consistency training (Geng et al., 2024) or distill from diffusion model (Song et al., 2023; Luo et al., 2023). CM (Song et al., 2023) is the first work proposing consistency distillation (CD) on pixel space. LCM (Luo et al., 2023) later applies consistency technique on latent space and can generate high-quality images within a few steps. However, LCM's generated images using 1-2 steps are still blurry (Luo et al., 2023). Recent works, such as Hyper-SD Ren et al. (2024) and TCD Zheng et al. (2024), have introduced notable improvements to latent consistency distillation. TCD Zheng et al. (2024) employed CTM Kim et al. (2023) instead of CD Song et al. (2023), significantly enhancing the performance of the distilled student model. Building on this, Hyper-SD Ren et al. (2024) divided the Probability Flow ODE (PF-ODE) into multiple components inspired by Multistep Consistency Models (MCM) Heek et al. (2024), and applied TCD Zheng et al. (2024) to each segment. Subsequently, Hyper-SD Ren et al. (2024) merged these segments progressively into a final model, integrating human feedback learning and score distillation Yin et al. (2024) to optimize one-step generation performance.

## 3 PRELIMINARIES

Denote $p_{\text{data}}(\mathbf{x}_0)$ as the data distribution, the forward diffusion process gradually adds Gaussian noise with monotonically increasing standard deviation $\sigma(t)$ for $t \in \{0, 1, \dots, T\}$ such that $p_t(\mathbf{x}_t|\mathbf{x}_0) = \mathcal{N}(\mathbf{x}_0, \sigma^2(t)\boldsymbol{I})$ and $\sigma(t)$ is handcrafted such that $\sigma(0) = \sigma_{\min}$ and $\sigma(T) = \sigma_{\max}$. By setting $\sigma(t) = t$, the probability flow ODE (PF-ODE) from (Karras et al., 2022) is defined as:

$$\frac{d\mathbf{x}_t}{dt} = -t\nabla_{\mathbf{x}_t} \log p_t(\mathbf{x}_t) = \frac{(\mathbf{x}_t - \boldsymbol{f}(\mathbf{x}_t, t))}{t}, \tag{1}$$

where $\boldsymbol{f} : (\mathbf{x}_t, t) \to \mathbf{x}_0$ is the denoising function which directly predicts clean data $\mathbf{x}_0$ from given perturbed data $\mathbf{x}_t$. (Song et al., 2023) defines consistency model based on PF-ODE in eq. (1), which builds a bijective mapping $\boldsymbol{f}$ between noisy distribution $p(\mathbf{x}_t)$ and data distribution $p_{\text{data}}(\mathbf{x}_0)$. The bijective mapping $\boldsymbol{f} : (\mathbf{x}_t, t) \to \mathbf{x}_0$ is termed the consistency function. A consistency model $\boldsymbol{f}_\theta(\mathbf{x}_t, t)$ is trained to approximate this consistency function $\boldsymbol{f}(\mathbf{x}_t, t)$. The previous works (Song et al., 2023; Song & Dhariwal, 2023; Karras et al., 2022) impose the boundary condition by parameterizing the consistency model as:

$$\boldsymbol{f}_\theta(\mathbf{x}_t, t) = c_{skip}(t)\mathbf{x}_t + c_{out}(t)\boldsymbol{F}_\theta(\mathbf{x}_t, t), \tag{2}$$

where $\boldsymbol{F}_\theta(\mathbf{x}_t, t)$ is a neural network to train. Note that, since $\sigma(t) = t$, we hereafter use $t$ and $\sigma$ interchangeably. $c_{skip}(t)$ and $c_{out}(t)$ are time-dependent functions such that $c_{skip}(\sigma_{\min}) = 1$ and $c_{out}(\sigma_{\max}) = 0$.

To train or distill consistency model, (Song et al., 2023; Song & Dhariwal, 2023; Karras et al., 2022) firstly discretize the PF-ODE using a sequence of noise levels $\sigma_{\min} = t_{\min} = t_1 < t_2 < \cdots < t_N = t_{\max} = \sigma_{\max}$, where $t_i = \left(t_{\min}^{1/\rho} + \frac{i-1}{N-1}(t_{\max}^{1/\rho} - t_{\min}^{1/\rho})\right)^\rho$ and $\rho = 7$.

**Consistency Distillation** Given the pretrained diffusion model $\boldsymbol{s}_\phi(\mathbf{x}_t, t) \approx \nabla_{\mathbf{x}_t} \log p_t(\mathbf{x}_t)$, the consistency model could be distilled from the pretrained diffusion model using the following CD loss:

$$\mathcal{L}_{\text{CD}}(\theta, \theta^-) = \mathbb{E}\left[\lambda(t_i)d(\boldsymbol{f}_\theta(\mathbf{x}_{t_{i+1}}, t_{i+1}), \boldsymbol{f}_{\theta^-}(\tilde{\mathbf{x}}_{t_i}, t_i))\right], \tag{3}$$

where $\mathbf{x}_{t_{i+1}} = \mathbf{x}_0 + t_{i+1}\mathbf{z}$ with the $\mathbf{x}_0 \sim p_{\text{data}}(\mathbf{x}_0)$ and $\mathbf{z} \sim \mathcal{N}(0, \boldsymbol{I})$ and $\mathbf{x}_{t_i} = \mathbf{x}_{t_{i+1}} - (t_i - t_{i+1})t_{i+1}\nabla_{\mathbf{x}_{t_{i+1}}} \log p_{t_{i+1}}(\mathbf{x}_{t_{i+1}}) = \mathbf{x}_{t_{i+1}} - (t_i - t_{i+1})t_{i+1}\boldsymbol{s}_\phi(\mathbf{x}_{t_{i+1}}, t_{i+1})$.

**Consistency Training** The consistency model is trained by minimizing the following CT loss:

$$\mathcal{L}_{\text{CT}}(\theta, \theta^-) = \mathbb{E}\left[\lambda(t_i)d(\boldsymbol{f}_\theta(\mathbf{x}_{t_{i+1}}, t_{i+1}), \boldsymbol{f}_{\theta^-}(\mathbf{x}_{t_i}, t_i))\right], \tag{4}$$

where $\mathbf{x}_{t_i} = \mathbf{x}_0 + t_i\mathbf{z}$ and $\mathbf{x}_{t_{i+1}} = \mathbf{x}_0 + t_{i+1}\mathbf{z}$ with the same $\mathbf{x}_0 \sim p_{\text{data}}(\mathbf{x}_0)$ and $\mathbf{z} \sim \mathcal{N}(0, \boldsymbol{I})$

In eq. (3) and eq. (4), $\boldsymbol{f}_\theta$ and $\boldsymbol{f}_{\theta^-}$ are referred to as the online network and the target network, respectively. The target's parameter $\theta^-$ is obtained by applying the Exponential Moving Average (EMA) to the student's parameter $\theta$ during the training and distillation as follows:

$$\theta^- \leftarrow \text{stopgrad}(\mu\theta^- + (1-\mu)\theta), \tag{5}$$

with $0 \leq \mu < 1$ as the EMA decay rate, weighting function $\lambda(t_i)$ for each timestep $t_i$, and $d(\cdot, \cdot)$ is a predefined metric function.

In CM (Song et al., 2023), the consistency training still lags behind the consistency distillation and diffusion models. iCT (Song & Dhariwal, 2023) later propose several improvements that significantly boost the training performance and efficiency. First, the EMA decay rate $\mu$ is set to 0 for better training convergence. Second, the Fourier scaling factor of noise embedding and the dropout rate are carefully examined. Third, iCT introduces Pseudo-Huber losses to replace $L_2$ and LPIPS since LPIPS introduces the undesirable bias in generative modeling (Song & Dhariwal, 2023). Furthermore, the Pseudo-Huber is more robust to outliers since it imposes a smaller penalty for larger errors than the $L_2$ metric. Fourth, iCT proposes an exp curriculum for total discretization steps N, which doubles N after a predefined number of training iterations. Moreover, uniform weighting $\lambda(t_i) = 1$ is replaced by $\lambda(t_i) = 1/(t_{i+1} - t_i)$. Finally, iCT adopts a discrete Lognormal distribution for timestep sampling as EDM (Karras et al., 2022). With all these improvements, CT is now better than CD and performs on par with the diffusion models in pixel space.

## 4  METHOD

In this paper, we first investigate the underlying reason behind the performance discrepancy between latent and pixel space using the same training framework in section 4.1. Based on the analysis, we find out the root of unsatisfied performance on latent space could be attributed to two factors: the impulsive outlier and the unstable temporal difference (TD) for computing consistency loss. To deal with impulsive outliers of TD on pixel space, (Song & Dhariwal, 2023) proposes the Pseudo-Huber function as training loss. For the latent space, the impulsive outlier is even more severe, making Pseudo-Huber loss not enough to resist the outlier. Therefore, section 4.2 introduces Cauchy loss, which is more effective with extreme outliers. In the next section 4.3 and section 4.4, we propose to use diffusion loss at early timesteps and OT matching for regularizing the overkill effect of consistency at the early step and training variance reduction, respectively. Section 4.5 designs an adaptive scheduler of scaling $c$ to control the robustness of the proposed loss function more carefully, leading to better performance. Finally, in section 4.6, we investigate the normalization layers of architecture and introduce Non-scaling LayerNorm to both capture feature statistic better and reduce the sensitivity to outliers.

### 4.1  ANALYSIS OF LATENT SPACE

We first reimplement the iCT model (Song & Dhariwal, 2023) on the latent dataset CelebA-HQ $32 \times 32 \times 4$ and pixel dataset CIFAR-10 $32 \times 32 \times 3$. Hereafter, we refer to the latent iCT model as iLCT. We find that iCT framework works well on pixel datasets as claim (Song & Dhariwal, 2023). However, it produces worse results on latent datasets as in fig. 5 and table 1. The iLCT gets a very high FID above 30 for both datasets, and the generative images are not usable in the real world. This observation raises concern about the sensitivity of CT algorithm with training data, and we should carefully examine the training dataset. In addition, we notice that the DQN and CM use the same TD loss, which update the current state using the future state. Furthermore, they also possess the training instability. This motivates to carefully examine the behavior of TD loss with different training data.

While the pixel data lies within the range $[-1, 1]$ after being normalized, the range of latent data varies depending on the encoder model, which is blackbox and unbound. After normalizing latent data using mean and variance, we observe that the latent data contains high-magnitude values. We call them the impulsive outliers since they account for small probability but are usually very large values. In the bottom left of fig. 1, the impulsive outlier of latent data is red, spanning from $-9$ to $7$, while the first and third quartiles are just around $-1.4$ and $1.4$, respectively. We evaluate how the iCT will be affected by data outliers by analyzing the temporal difference TD $= f_\theta(\mathbf{x}_{t_{i+1}}, t_{i+1}) - f_{\theta^-}(\mathbf{x}_{t_i}, t_i)$. In the top right of fig. 1, the impulsive outliers of pixel TD range from -1.5 to 1.7, which are not too far from the interquartile range compared to latent TD. The impulsive outliers of latent TD range is much wider from -3.2 to 5. iCT uses Pseudo-Huber loss instead of $L_2$ loss since the Huber is less sensitive to outliers, see fig. 2. However, for latent data, the Huber's reduction in sensitivity to outliers is not enough. This indicates that even using Pseudo-Huber loss, the iLCT training on latent space could still be unstable and lead to worse performance, which matches our

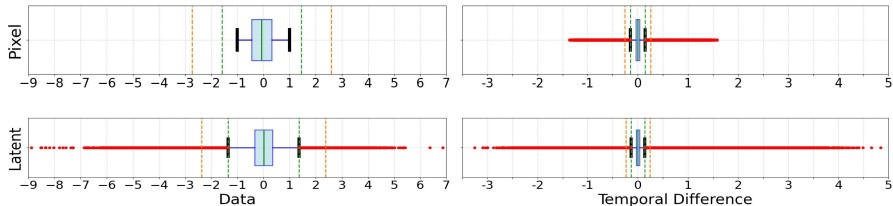

Figure 1: **Box and Whisker Plot:** Impulsive noise comparison between pixel and latent spaces. The right column shows the statistics of TD values at 21 discretization steps. Other discretization steps exhibit same behavior, where impulsive outliers are consistently present regardless of the total discretization steps. The blue boxes represent interquartile ranges of the data, while the green and orange dashed lines indicate inner and outer fences, respectively. Outliers are marked with red dots.

experiment results on iLCT. Based on the above analysis, we hypothesize that the TD value statistic highly depends on the training data statistic.

To mitigate the impact of impulsive outliers, we could use more stable target updates like Polyak or periodic in TD loss Lee & He (2019), but they lead to very slow convergence, as shown in (Song et al., 2023). Even though CM is initialized by a pretrained diffusion model, the Polyak update still takes a long time to converge. Therefore, using Polyak or periodic updates is computationally expensive, and we keep the standard target update as in (Song & Dhariwal, 2023). Another direction is using a special metric for latent like LPIPS on pixel space (Song et al., 2023). (Kang et al., 2024) proposes the E-LatentLPIPS as a metric for distillation and performs well on distillation tasks. However, this requires training a network as a metric and using this metric during the training process will also increase the training budget. To avoid the overhead of the training, we seek a simple loss function like Pseudo-Huber but be more effective with outliers. We find that the Cauchy loss function (Black & Anandan, 1996; Barron, 2019) could be a promising candidate in place of Pseudo-Huber for latent space.

## 4.2 CAUCHY LOSS AGAINST IMPULSIVE OUTLIER

In this section, we introduce the Cauchy loss (Black & Anandan, 1996; Barron, 2019) function to deal with extreme impulsive outliers. The Cauchy loss function has the following form:

$$d_{\text{Cauchy}}(\mathbf{x}, \mathbf{y}) = \log\left(1 + \frac{||\mathbf{x} - \mathbf{y}||_2^2}{2c^2}\right),\tag{6}$$

and we also consider two additional robust losses, which are Pseudo-Huber (Song & Dhariwal, 2023; Barron, 2019) and Geman-McClure (Geman & Geman, 1986; Barron, 2019)

$$d_{\text{Pseudo-Huber}}(\mathbf{x}, \mathbf{y}) = \sqrt{||\mathbf{x} - \mathbf{y}||_2^2 + c^2} - c,\tag{7}$$

$$d_{\text{Geman-McClure}}(\mathbf{x}, \mathbf{y}) = \frac{2||\mathbf{x} - \mathbf{y}||_2^2}{||\mathbf{x} - \mathbf{y}||_2^2 + 4c^2},\tag{8}$$

where $c$ is the scaling parameter to control how robust the loss is to the outlier. We analyze their robustness behavior against outliers. As shown in fig. 2a, the Pseudo-Huber loss linearly increases like $L_1$ loss for the large residuals $\mathbf{x} - \mathbf{y}$. In contrast, the Cauchy loss only grows logarithmically, and the Geman-McClure suppresses the loss value to 1 for the outliers.

The Pseudo-Huber loss works well if the residual value does not grow too high and, therefore, has a good performance on the pixel space. However, for the latent space, as shown in the bottom right of fig. 1, the TD suffers from extremely high values coming from the impulsive outlier in the latent dataset, the Cauchy loss could be more suitable since it significantly dampens the influence of extreme outliers. Otherwise, even Geman-McClure is very highly effective for removing outlier effects than two previous losses; it gives a gradient 0 for high TD value and completely ignores the impulsive outliers as fig. 2b. This is unexpected behavior because even though we call the high-value latent impulsive outlier, they actually could encode important information from original data. Completely ignoring them could significantly hurt the performance of training model. Based on this analysis, we choose Cauchy loss as the default loss for latent CM for the rest of the paper. The loss ablation is provided in table 2c.

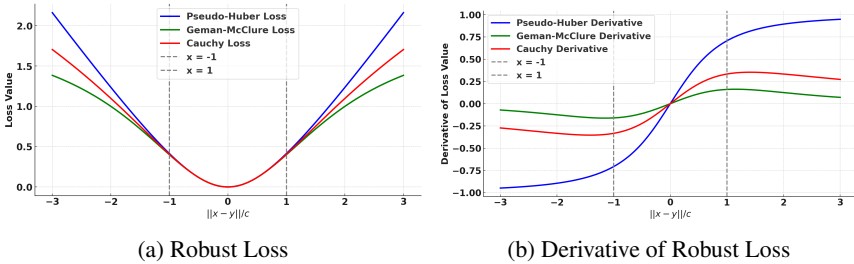

(a) Robust Loss          (b) Derivative of Robust Loss

Figure 2: Analysis of robust loss: Pseudo-Huber, Cauchy, and Geman-McClure

### 4.3 DIFFUSION LOSS AT SMALL TIMESTEP

For small noise level $\sigma$, the ground truth of $f(\mathbf{x}_\sigma, \sigma)$ can be well approximated by $\mathbf{x}_0$, but this does not hold for large noise levels. Therefore, for low-level noise, the consistency objective seems to be overkill and harms the model's performance since instead of optimizing $f_\theta(\mathbf{x}_\sigma, \sigma)$ to approximated ground truth $\mathbf{x}_0$, the consistency objective optimizes through a proxy estimator $f_{\theta^-}(\mathbf{x}_{<\sigma}, < \sigma)$ leading to error accumulation over timestep. To regularize this overkill, we propose to apply an additional diffusion loss on small noise level as follows:

$$L_{diff} = ||f_\theta(\mathbf{x}_{t_i}, t_i) - \mathbf{x}_0||_2^2 \quad \forall i \leq \text{int}(\text{N} \cdot \text{r}), \tag{9}$$

where N is the number of training discretization steps and $r \in [0; 1]$ is the diffusion threshold, and we heuristicly choose $r = 0.25$. We do not apply diffusion loss for large noise levels since $f(\mathbf{x}_\sigma, \sigma)$ will differ greatly from the target $\mathbf{x}_0$, leading to very high $L_2$ diffusion loss. This could harm the training consistency process, misleading to the wrong solution. We provide the ablation study in table 2b. Furthermore, CTM (Kim et al., 2023) also proposes to use diffusion loss, but they use them on both high and low-level noise, which is different from us.

### 4.4 OT MATCHING REDUCES THE VARIANCE

In this section, we adopt the OT matching technique from previous works (Pooladian et al., 2023; Lee et al., 2023). (Pooladian et al., 2023) proposes to use OT to match noise and data in the training batch, such as the moving $L_2$ cost is optimal. On the other hand, (Lee et al., 2023) introduces $\beta$VAE for creating noise corresponding to data and train flow matching on the defined data-noise pairs. By reassigning noise-data pairs, these works significantly reduce the variance during the diffusion/flow matching training process, leading to a faster and more stable training process. According to (Zhang et al., 2023a), the consistency training and diffusion models produce highly similar images given the same noise input. Therefore, the final output solution of the consistency and diffusion models should be close to each other. Since OT matching helps reduce the variance during training diffusion, it could be useful to reduce the variance of consistency training. In our implementation, we follow (Pooladian et al., 2023; Tong et al., 2023) using the POT library to map from noise to data in the training batch. The overhead caused by minibatch OT is relatively small, only around $0.93\%$ training time, but gains significant performance improvement as shown in table 2a.

### 4.5 ADAPTIVE $c$ SCHEDULER

In this section, we examine the choice of scaling parameter $c$ in robust loss functions. The scaling parameter controls the robustness level, which is very important for model performance. The previous work (Song & Dhariwal, 2023) proposes to use fixed constant $c_0 = 0.00054\sqrt{d}$, where $d$ is the dimension of data. We find that using this simple fixed $c$ is not yet optimal for the training consistency model. Especially in this paper, we follow the Exp curriculum specified by eq. (10) in (Song & Dhariwal, 2023), which doubles the total discretization step after a defined number of training iterations.

$$\text{NFE}(k) = \min\left(s_0 2^{\lfloor \frac{k}{K'} \rfloor}, s_1\right) + 1, \quad K' = \left\lfloor \frac{K}{\log_2 \lfloor s_1/s_0 \rfloor + 1} \right\rfloor, \tag{10}$$

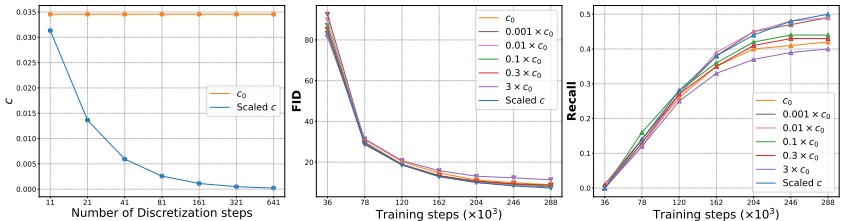

Figure 3: Model convergence plot on different $c$ schedule. (Left) Our proposed $c$ values. Performance on FID (Middle) and Recall (Right) of our proposed $c$ in comparison with different choices.

where $k$ is current training iteration, $K$ is total training iteration and $s_0 = 10, s_1 = 640$. During training, we notice that the variance of TD is significantly reduced as doubling total discretization steps using eq. (10). Since the more discretization steps, the closer distance of $\mathbf{x}_{t_i}$ and $\mathbf{x}_{t_{i+1}}$, the TD value's range between them should be smaller. However, the impulsive outlier still exists regardless of the number of discretization steps. Intuitively, we propose a heuristic adaptive $c$ scheduler where the $c$ is scaled down proportional to the reduction rate of TD variance as the number of discretization steps increases. We plot our $c$ scheduler versus discretization steps in fig. 3 and we fit the $c$ scheduler to get the scheduler equation as following:

$$c = \exp(-1.18 * \log(\text{NFE}(k) - 1) - 0.72) \tag{11}$$

### 4.6 NON-SCALING LAYERNORM

As mentioned in section 4.1, the statistic of training data could play an important role in the success of consistency training. Furthermore, in architecture design, the normalization layer specifically handles the statistics of input, output, and hidden features. In this section, we investigate the normalization layer choice for consistency training, which is sensitive to training data statistics.

Currently, both (Song & Dhariwal, 2023; Song et al., 2023) use the UNet architecture from (Dhariwal & Nichol, 2021). In UNet (Dhariwal & Nichol, 2021), GroupNorm is used in every layer by default. The GroupNorm only captures the statistics over groups of local channels, while the LayerNorm further captures the statistics' overall features. Therefore, LayerNorm is better at capturing fine-grained statistics over the entire feature. We further carry out the experiments for other types of normalization, such as LayerNorm, InstanceNorm, RMSNorm in table 2d and observe that the GroupNorm and InstanceNorm perform relatively well compared to others, especially LayerNorm. This could be due to that they are less sensitive to the outliers since they only capture the statistic over groups of channels. Therefore, the impulsive features only affect the normalization of a group containing them. For the LayerNorm, the impulsive features could negatively impact the overall features's normalization. We further look into the LayerNorm implementation and suspect that the scaling term could significantly amplify the outliers across features by serving as a shared parameter. This observation is also mentioned in (Wei et al., 2022) for LLM quantization. In implementation, we set the **scaling term of LayerNorm to** 1 and **disabled the gradient update** for it equation 12. We refer to it as Non-scaling LayerNorm (NsLN) as (Wei et al., 2022).

$$\text{LN}_{\gamma,\beta}(\mathbf{x}) = \frac{\mathbf{x} - u(\mathbf{x})}{\sqrt{\sigma^2(\mathbf{x}) + \epsilon}} \cdot \gamma + \beta, \quad \text{NsLN}_\beta(\mathbf{x}) = \frac{\mathbf{x} - u(\mathbf{x})}{\sqrt{\sigma^2(\mathbf{x}) + \epsilon}} + \beta, \tag{12}$$

where $u(\mathbf{x})$ and $\sigma^2(\mathbf{x})$ are mean and variance of $\mathbf{x}$.

## 5 EXPERIMENT

### 5.1 PERFORMANCE OF OUR TRAINING TECHNIQUE

**Experiment Setting:** We measure the performance of our proposed technique on three datasets: CelebA-HQ (Huang et al., 2018), FFHQ (Karras et al., 2019), and LSUN Church (Yu et al., 2015),

| Model | NFE↓ | FID↓ | Recall↑ | Epochs | Total Bs |
|---|---|---|---|---|---|
| **Pixel Diffusion Model** | | | | | |
| WaveDiff (Phung et al., 2023) | 2 | 5.94 | 0.37 | 500 | 64 |
| Score SDE (Song et al., 2020) | 4000 | 7.23 | - | 6.2K | - |
| DDGAN (Xiao et al., 2021) | 2 | 7.64 | 0.36 | 800 | 32 |
| RDUOT (Dao et al., 2024b) | 2 | 5.60 | 0.38 | 600 | 24 |
| RDM (Teng et al., 2023) | 270 | 3.15 | 0.55 | 4K | - |
| UNCSN++ (Kim et al., 2021) | 2000 | 7.16 | - | - | - |
| **Latent Diffusion Model** | | | | | |
| LFM-8 (Dao et al., 2023) | 85 | 5.82 | 0.41 | 500 | 112 |
| LDM-4 (Rombach et al., 2021) | 200 | 5.11 | 0.49 | 600 | 48 |
| LSGM (Vahdat et al., 2021) | 23 | 7.22 | - | 1K | - |
| DDMI (Park et al., 2024) | 1000 | 7.25 | - | - | - |
| DIMSUM (Phung et al., 2024) | 73 | 3.76 | 0.56 | 395 | 32 |
| LDM-8[†] | 250 | 8.85 | - | 1.4K | 128 |
| **Latent Consistency Model** | | | | | |
| iLCT (Song & Dhariwal, 2023) | 1 | 37.15 | 0.12 | 1.4K | 128 |
| iLCT (Song & Dhariwal, 2023) | 2 | 16.84 | 0.24 | 1.4K | 128 |
| Ours | 1 | 7.27 | 0.50 | 1.4K | 128 |
| Ours | 2 | 6.93 | 0.52 | 1.4K | 128 |

(a) CelebA-HQ

| Model | NFE↓ | FID↓ | Recall↑ | Epochs | Total Bs |
|---|---|---|---|---|---|
| **Pixel Diffusion Model** | | | | | |
| WaveDiff (Phung et al., 2023) | 2 | 5.94 | 0.37 | 500 | 64 |
| Score SDE (Song et al., 2020) | 4000 | 7.23 | - | 6.2K | - |
| DDGAN (Xiao et al., 2021) | 2 | 5.25 | 0.36 | 500 | 32 |
| **Latent Diffusion Model** | | | | | |
| LFM-8 (Dao et al., 2023) | 90 | 7.70 | 0.39 | 90 | 112 |
| LDM-8 (Rombach et al., 2021) | 400 | 4.02 | 0.52 | 400 | 96 |
| LDM-8[†] | 250 | 10.81 | - | 1.8K | 256 |
| **Latent Consistency Model** | | | | | |
| iLCT (Song & Dhariwal, 2023) | 1 | 52.45 | 0.11 | 1.8K | 256 |
| iLCT (Song & Dhariwal, 2023) | 2 | 24.67 | 0.17 | 1.8K | 256 |
| Ours | 1 | 8.87 | 0.47 | 1.8K | 256 |
| Ours | 2 | 7.71 | 0.48 | 1.8K | 256 |

(b) LSUN Church

| Model | NFE↓ | FID↓ | Recall↑ | Epochs | Total Bs |
|---|---|---|---|---|---|
| **Latent Diffusion Model** | | | | | |
| LFM-8 (Dao et al., 2023) | 84 | 8.07 | 0.40 | 700 | 128 |
| LDM-4 (Rombach et al., 2021) | 200 | 4.98 | 0.50 | 400 | 42 |
| LDM-8[†] | 250 | 10.23 | - | 1.4K | 128 |
| **Latent Consistency Model** | | | | | |
| iLCT (Song & Dhariwal, 2023) | 1 | 48.82 | 0.15 | 1.4K | 128 |
| iLCT (Song & Dhariwal, 2023) | 2 | 21.15 | 0.19 | 1.4K | 128 |
| Ours | 1 | 8.72 | 0.42 | 1.4K | 128 |
| Ours | 2 | 8.29 | 0.43 | 1.4K | 128 |

(c) FFHQ

Table 1: Our performance on CelebA-HQ, LSUN Church, FFHQ datasets at resolution $256 \times 256$. (†) means training on our machine with the same diffusion forward and equivalent architecture.

at the same resolution of $256 \times 256$. Following LDM (Rombach et al., 2021), we use pretrained VAE KL-8 [†] to obtain latent data with the dimensionality of $32 \times 32 \times 4$. We adopt the OpenAI UNet architecture (Dhariwal & Nichol, 2021) as the default architecture throughout the paper. Furthermore, we use the variance exploding (VE) forward process for all the consistency and diffusion experiments following (Song et al., 2023; Song & Dhariwal, 2023). The baseline iCT is self-implemented based on official implementation CM (Song et al., 2023) and iCT (Song & Dhariwal, 2023). We refer to this baseline as iLCT. Furthermore, we also train the latent diffusion model for each dataset using the same VE forward noise process for fair comparisons with our technique. This LDM model is referred to as LDM-8[†] in table 1. All three frameworks, including ours, iLCT, and LDM-8[†], use the same architecture.

**Evaluation:** During the evaluation, we first generate 50K latent samples and then pass them through VAE's decoder to obtain the pixel images. We use two well-known metrics, Fréchet Inception Distance (FID) (Naeem et al., 2020) and Recall (Kynkäänniemi et al., 2019), for measuring the performance of the model given the training data and 50K generated images.

**Model Performance:** We report the performance of our model across all three datasets in table 1, primarily to compare it with the baseline iLCT (Song & Dhariwal, 2023) and LDM (Rombach et al., 2021). For both 1 and 2 NFE sampling, we observe that the FIDs of iLCT for all datasets are notably high (over 30 for 1-NFE sampling and over 16 for 2-NFE sampling), consistent with the qualitative results shown in fig. 5, where the generated image is unrealistic and contain many artifacts. This poor performance of iLCT in latent space is expected, as the Pseudo-Huber training losses are insufficient in mitigating extreme impulsive outliers, as discussed in section 4.1 and section 4.2. In contrast, our proposed framework demonstrates significantly better FID and Recall than iLCT. Specifically, we achieve 1-NFE sampling FIDs of 7.27, 8.87, and 8.29 for CelebA-HQ, LSUN Church, and FFHQ, respectively. For 2-NFE sampling, our FID scores improve across all three datasets. Notably, our 1-NFE sampling outperforms LDM-8[†], using the same noise scheduler and architecture. However, our models still exhibit higher FIDs compared to LDM (Rombach et al., 2021) and LFM (Dao et al., 2023). In contrast, we only need 1 or 2 timestep sampling, whereas they require multiple timesteps for high-fidelity generation. It's important to note that we employ the VE forward process, whereas these other methods use VP and flow-matching forward processes. Furthermore, the qualitative results of our framework, as shown in fig. 4, highlight our ability to generate high-quality images.

---

[†] https://huggingface.co/stabilityai/sd-vae-ft-ema

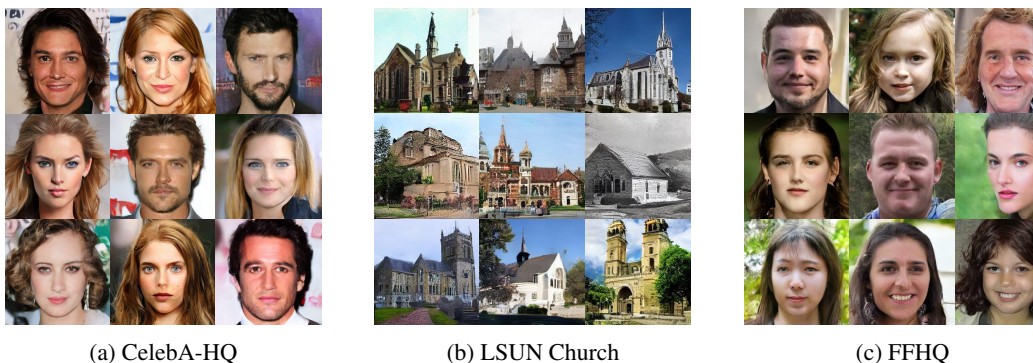

(a) CelebA-HQ            (b) LSUN Church            (c) FFHQ

Figure 4: Our qualitative results using 1-NFE at resolution $256 \times 256$

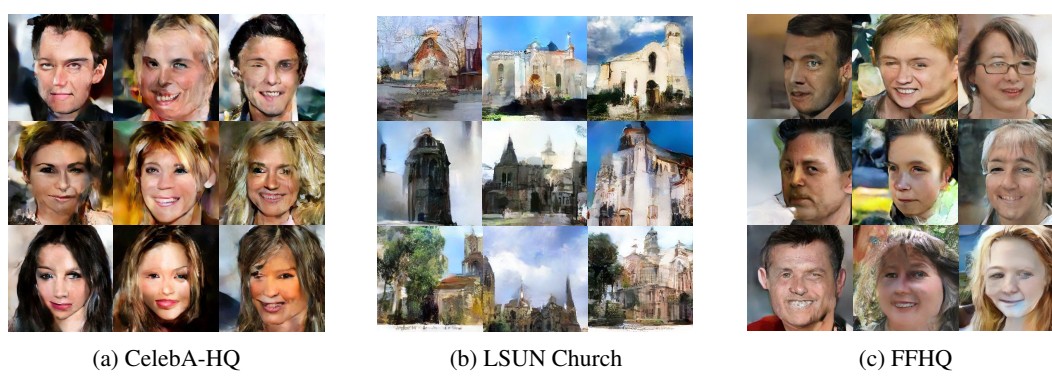

(a) CelebA-HQ            (b) LSUN Church            (c) FFHQ

Figure 5: iLCT qualitative results using 1-NFE at resolution $256 \times 256$

## 5.2 ABLATION OF PROPOSED FRAMEWORK

We ablate our proposed techniques on the CelebA-HQ $256 \times 256$ dataset, with all FID and Recall metrics measured using 1-NFE sampling. All models are trained for 1,400 epochs with the same hyperparameters. As shown in table 2a, replacing Pseudo-Huber losses with Cauchy losses makes our model's training less sensitive to impulsive outliers, resulting in a significant FID reduction from 37.15 to 13.02. This demonstrates the effectiveness of Cauchy losses in handling extremely high-value outliers, as discussed in section 4.2. Additionally, applying diffusion loss at small timesteps further reduces FID by approximately 4 points to 9.11, as this loss term stabilizes the training process at small timesteps, as described in section 4.3. Introducing OT coupling during minibatch training reduces training variance, improving the FID to 8.89. Notably, by replacing the fixed scaling term $c = c_0$, (Song & Dhariwal, 2023) with an adaptive scaling schedule, our model achieves an additional FID reduction of more than 1 point, reaching 7.76, highlighting the importance of the scaling term $c$ in robustness control. Finally, we propose using NsLN, which removes the scaling term from LayerNorm to handle outliers more effectively. NsLN captures feature statistics while mitigating the negative impact of outliers, resulting in our best FID of 7.27.

**Robustness Loss** To analyze the impact of different robust loss functions, we conduct an ablation study using our best settings but replace the Cauchy loss with alternatives such as L2, E-LatentLPIPS Kang et al. (2024), the Huber and the Geman-McClure loss. The results, shown in table 2c, indicate that both Huber and Geman-McClure underperform compared to the Cauchy loss when applied in the latent space. This is because the Huber loss remains too sensitive to extremely impulsive outliers, while the Geman-McClure loss tends to ignore such outliers entirely, leading to a loss of important information. This behavior is also discussed in section 4.2.

**Diffusion Threshold** In this section, we explore the impact of varying the threshold for applying the diffusion loss function in combination with the consistency loss. We observe that using the diffusion loss at every timestep improves consistency training; however, it underperforms compared

| Framework | FID $\downarrow$ | Recall $\uparrow$ |
|---|---|---|
| iLCT | 37.15 | 0.12 |
| Cauchy | 13.02 | 0.36 |
| + Diff | 9.11 | 0.41 |
| + OT | 8.89 | 0.42 |
| + Scaled $c$ | 7.76 | 0.47 |
| + NsLN | **7.27** | **0.50** |

(a) Components of proposed framework

| $r$ | FID $\downarrow$ | Recall $\uparrow$ |
|---|---|---|
| 1.0 | 7.47 | 0.49 |
| 0.6 | 7.33 | 0.49 |
| 0.25 | **7.27** | **0.50** |

(b) Threshold using Diffusion loss

| Loss | FID $\downarrow$ | Recall $\uparrow$ |
|---|---|---|
| L2 | 50.40 | 0.04 |
| E-LatentLPIPS | 11.49 | 0.47 |
| Huber | 9.97 | 0.44 |
| Geman McClure | 11.28 | 0.44 |
| Cauchy | **7.27** | **0.50** |

(c) Robust losses.

| Norm layer | FID $\downarrow$ | Recall $\uparrow$ |
|---|---|---|
| GN | 7.76 | 0.47 |
| IN | 8.47 | 0.43 |
| LN | 9.05 | 0.46 |
| RMS | 8.96 | 0.46 |
| NsLN | **7.27** | **0.50** |

(d) Norm Layer

Table 2: Ablation Studies on CelebA-HQ $256 \times 256$ dataset at epoch 1400

to applying the diffusion loss selectively at smaller timesteps such as $r = 0.25$ as shown in table 2b. This suggests that applying diffusion losses primarily at small noise levels improves performance as discussed section 4.3. At larger timesteps, the diffusion loss may conflict with the consistency loss, potentially guiding the model toward incorrect solutions, thereby reducing overall performance.

**Scaling term $c$ scheduler** In this section, we compare the performance of our adaptive scaling $c$ scheduler with the fixed scaling $c$ scheduler proposed in (Song & Dhariwal, 2023). Our model demonstrates better convergence with the proposed adaptive $c$ scheduler. The rationale behind this improvement lies in the fact that, as the discretization steps increases using the exponential curriculum, the value of the TD scales down. Despite the reduced TD value, impulsive outliers still persist. A fixed large scaling $c$ is not effective in handling these outliers. To address this, we scale $c$ down as discretization steps increases, which leads to better performance, as shown in fig. 3.

**Normalizing Layer** We denote GN, IN, LN, RMS, and NsLN as GroupNorm, InstanceNorm, LayerNorm, RMSNorm, and Non-scaling LayerNorm, respectively. The baseline UNet architecture from (Dhariwal & Nichol, 2021) uses GroupNorm by default. We replace the normalization layers in the baseline with each of these types and train the model on CelebA-HQ using the best settings. The results are reported in table 2d. GN and IN only capture local statistics, making them more robust to outliers, as outliers in one region do not affect others. In contrast, LN captures statistics from all features, making it more vulnerable to outliers because an outlier affects all features through a shared scaling term. By removing the scaling term in LN, we obtain NsLN, which is both effective in capturing feature statistics and resistant to outliers. As shown in table 2d, NsLN outperforms the second-best GN by 0.5 FID and significantly outperforms LN.

# 6 CONCLUSION

CT is highly sensitive to the statistical properties of the training data. In particular, when the data contains impulsive noise, such as latent data, CT becomes unstable, leading to poor performance. In this work, we propose using the Cauchy loss, which is more robust to outliers, along with several improved training strategies to enhance model performance. As a result, we can generate high-fidelity images from latent CT, effectively bridging the gap between latent diffusion models and consistency models. Future work could explore further improvements to the architecture, specifically by investigating normalization methods that reduce the impact of outliers. For example, removing the scaling term from group normalization or instance normalization may help mitigate outlier effects. Another promising future direction is the integration of this technique with Consistency Trajectory Models (CTM) Kim et al. (2023), as CTM has demonstrated improved performance compared to traditional Consistency Models (CM) Song et al. (2023).

ACKNOWLEDGEMENTS

Research funded by research grants to Prof. Dimitris Metaxas from NSF: 2310966, 2235405, 2212301, 2003874, 1951890, AFOSR 23RT0630, and NIH 2R01HL127661.

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

## A  APPENDIX

We provide additional uncurated samples of our models for three datasets: CelebaA-HQ (6, 7), LSUN Church (8, 9), and FFHQ (10, 11). We also provide additional uncurated samples of our models on CelebaA-HQ trained with L2 loss (12) and E-LatentLPIPS loss (13).

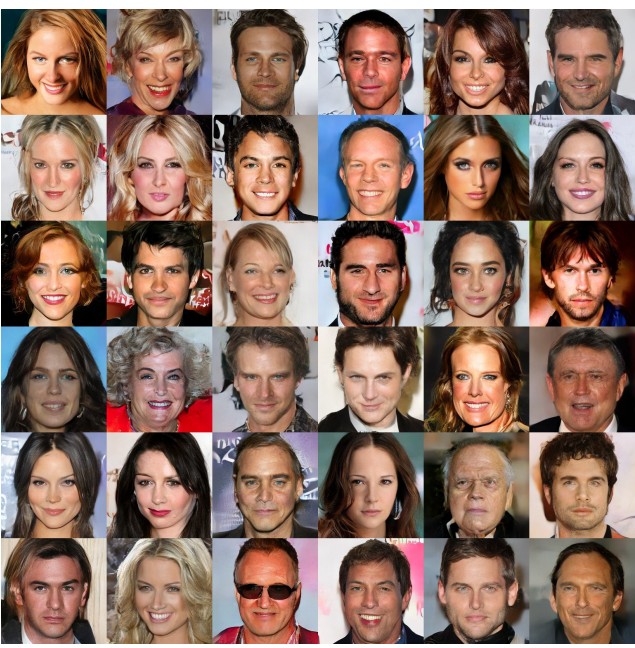

Figure 6: One-step samples on CelebA-HQ $256 \times 256$

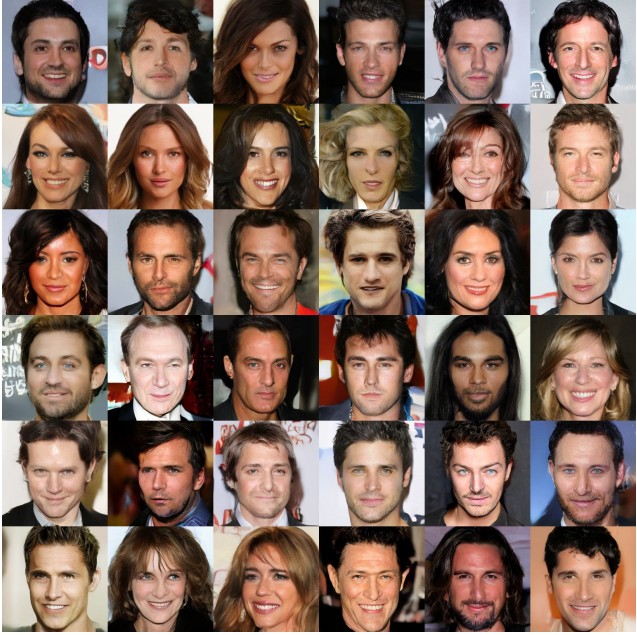

Figure 7: Two-step samples on CelebA-HQ $256 \times 256$

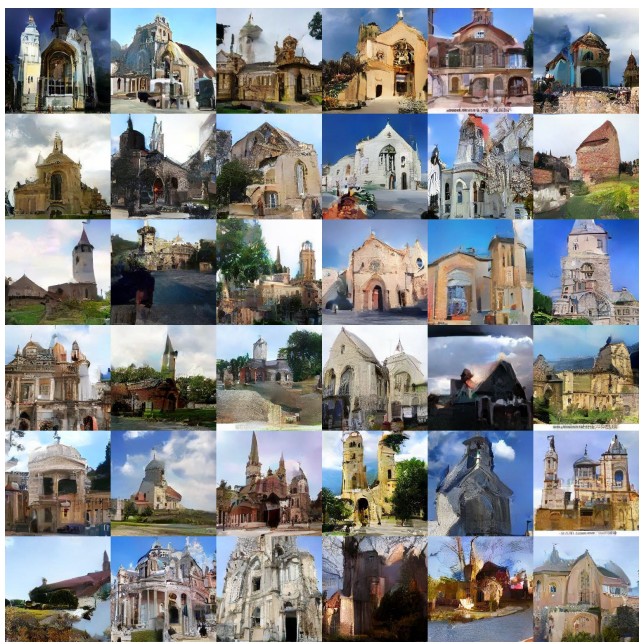

Figure 8: One-step samples on LSUN Church $256 \times 256$

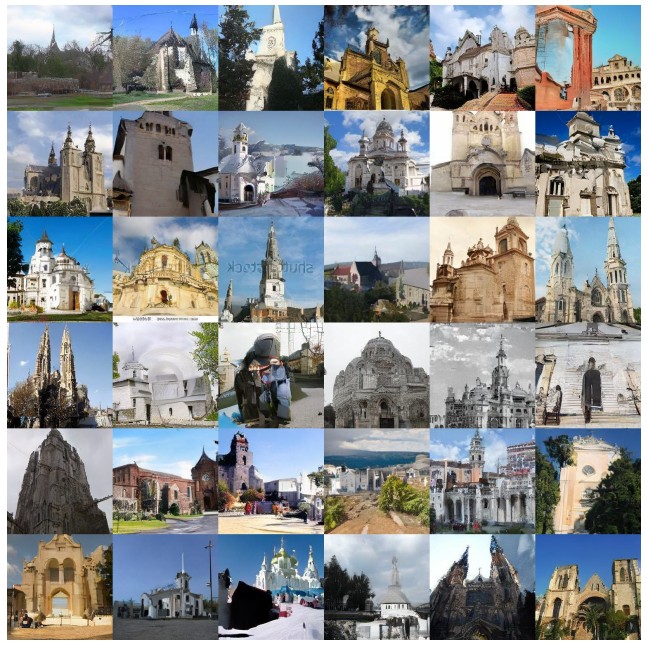

Figure 9: Two-step samples on LSUN Church $256 \times 256$

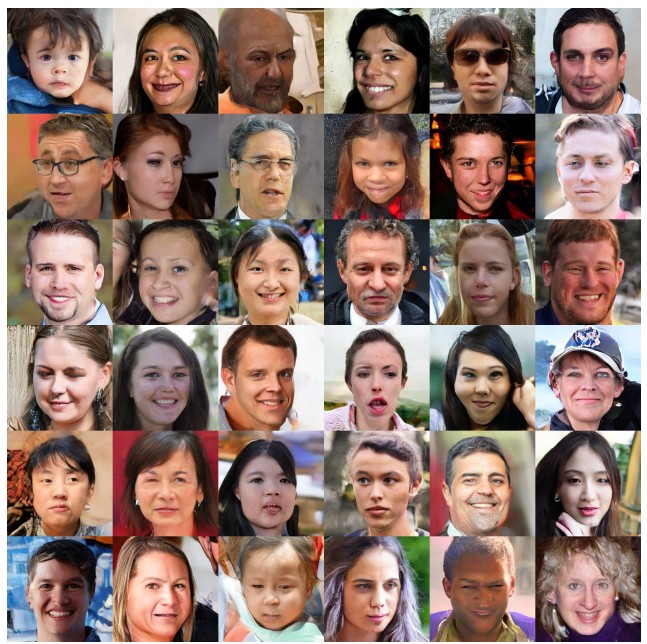

Figure 10: One-step samples on FFHQ $256 \times 256$

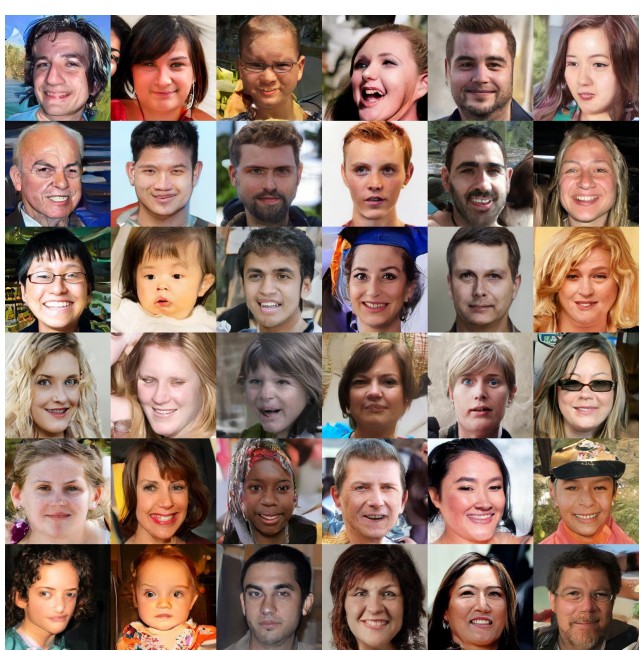

Figure 11: Two-step samples on FFHQ $256 \times 256$

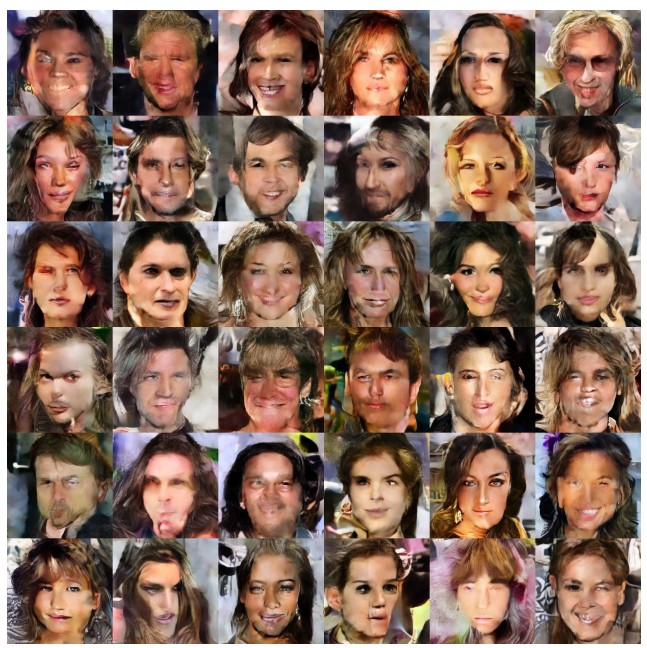

Figure 12: One-step samples on CelebA-HQ $256 \times 256$ (L2 loss)

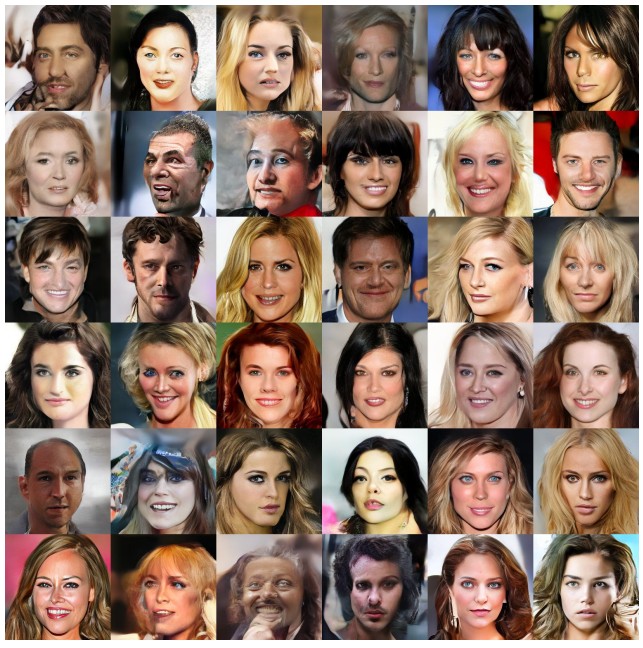

Figure 13: One-step samples on CelebA-HQ $256 \times 256$ (E-LatentLPIPS loss)

