# A   APPENDIX

We provide additional uncurated samples of our models for three datasets: CelebaA-HQ (6, 7), LSUN Church (8, 9), and FFHQ (10, 11). We also provide additional uncurated samples of our models on CelebaA-HQ trained with L2 loss (12) and E-LatentLPIPS loss (13).

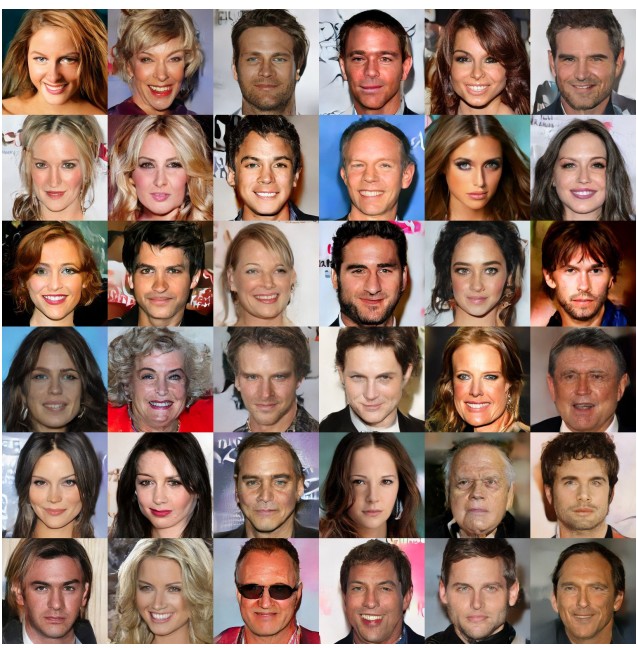

Figure 6: One-step samples on CelebA-HQ $256 \times 256$

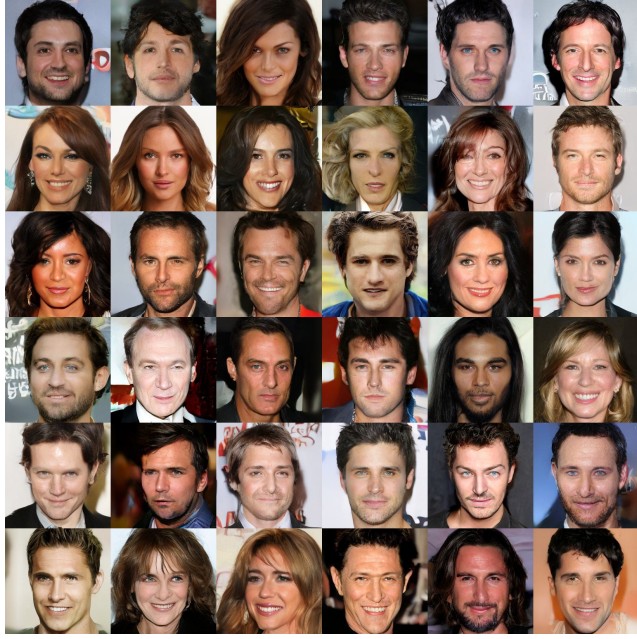

Figure 7: Two-step samples on CelebA-HQ $256 \times 256$

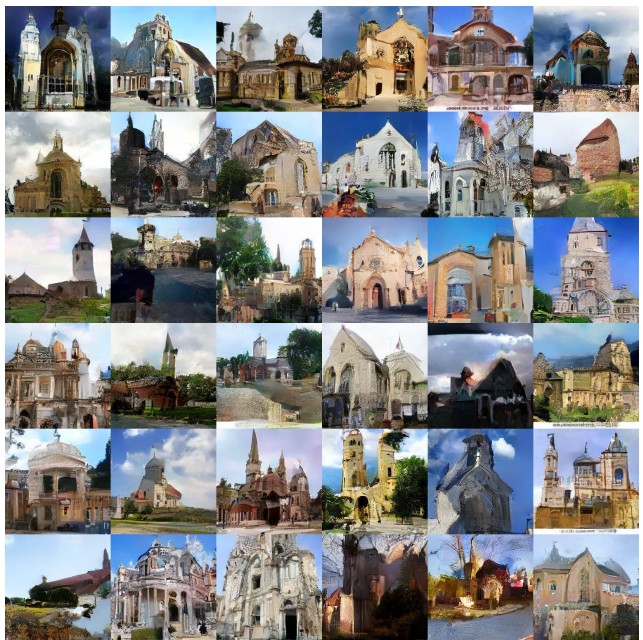

Figure 8: One-step samples on LSUN Church $256 \times 256$

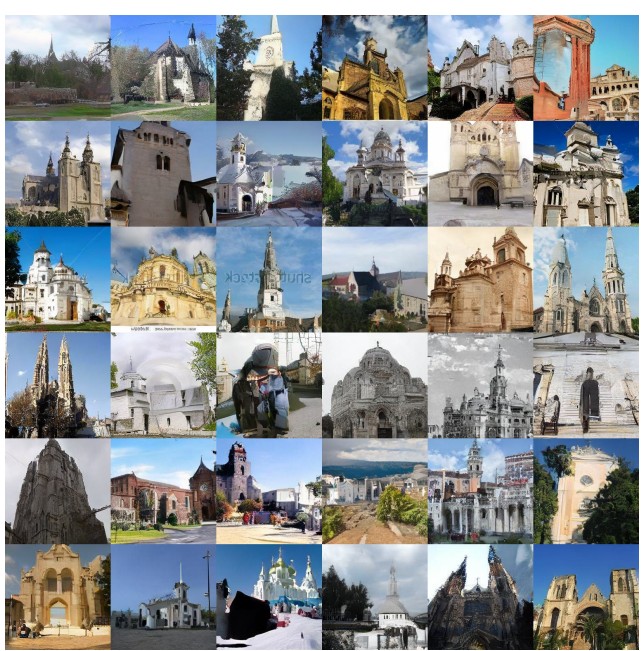

Figure 9: Two-step samples on LSUN Church $256 \times 256$

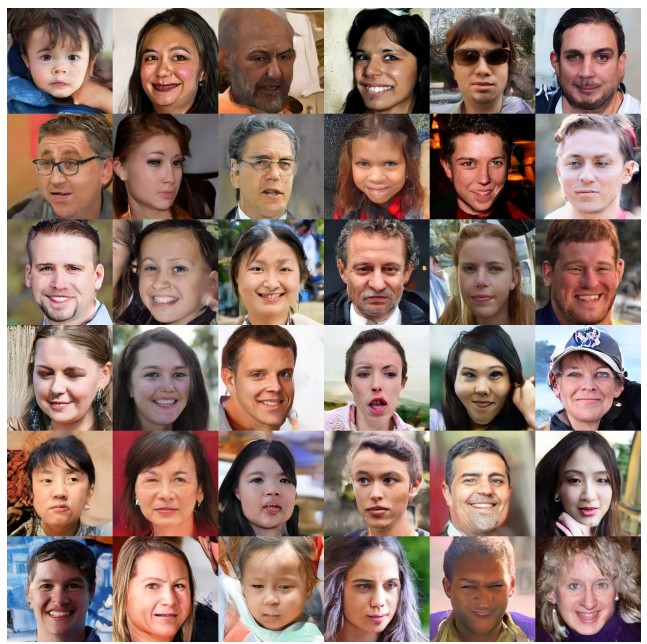

Figure 10: One-step samples on FFHQ $256 \times 256$

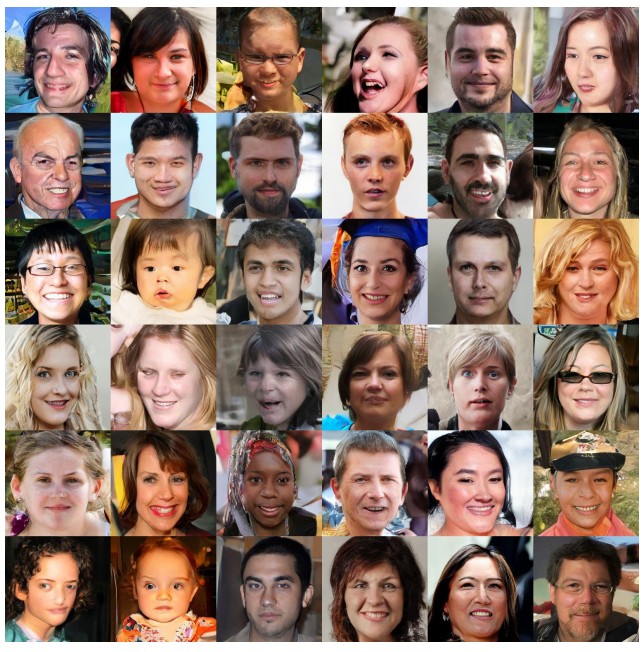

Figure 11: Two-step samples on FFHQ $256 \times 256$

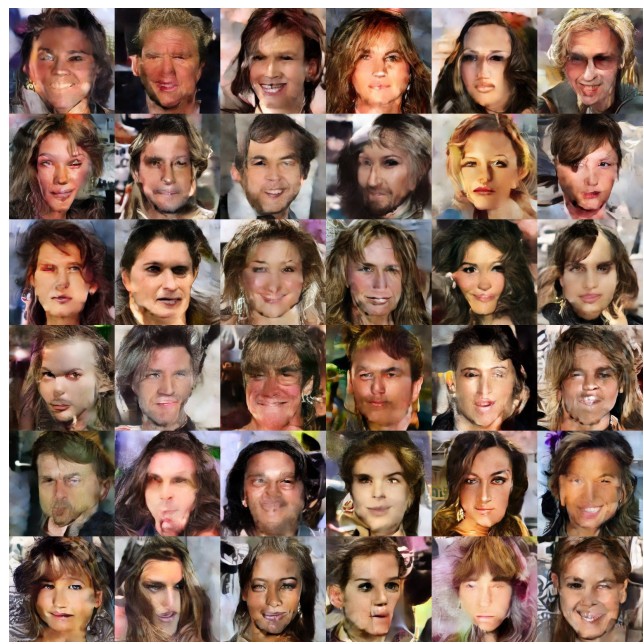

Figure 12: One-step samples on CelebA-HQ $256 \times 256$ (L2 loss)

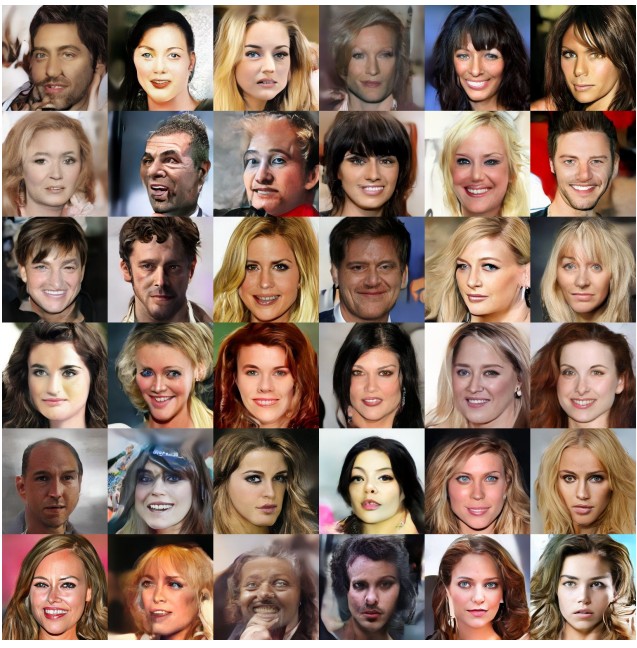

Figure 13: One-step samples on CelebA-HQ $256 \times 256$ (E-LatentLPIPS loss)