# OpenReview forum: "Improved Training Technique for Latent Consistency Models"
_ICLR.cc/2025/Conference — ICLR 2025 Poster_

### Official Review · Reviewer_n8Mw · 2024-10-31

**Soundness:** 2
**Presentation:** 2
**Contribution:** 2
**Rating:** 6
**Confidence:** 4

**Summary:**

This paper targets the efficient training problem for consistency models on latent space. They find that the highly impulsive outliers in the latent data significantly degrade the performance of iCT. Therefore, they employ a series of training tricks, including Cauchy loss, diffusion loss at early timesteps, optimal transport, adaptive scaling-c scheduler, and non-scaling layernorm to improve the performance.

**Strengths:**

1. The performance reported in the paper is good compared to the self-implemented iCT.
2. The motivation that delivers Cauchy loss, optimal transport, adaptive scaling-c scheduler, and non-scaling layernorm to handle highly impulsive outliers is convincing.

**Weaknesses:**

1. The soundness of the contribution in this paper is not good. The major contributions (Cauchy loss, optimal transport, adaptive scaling-c scheduler, and non-scaling layernorm) are mostly engineering training tricks. The correlation between the major contributions is not so strong, which makes the entire paper look like an A+B+C work. Since CM is already a complex model with so many hyper-parameters and training tricks, introducing more training tricks into CM doesn't appeal to me enough.

2. The comparison of this paper with SOTA models is not sufficient. While the authors mention LCM in the paper, they do not compare their model with LCM in Table 1. This paper is mainly based on the impulsive outliers observed from their self-implemented iLCT, which is not so convincing to me. Does LCM also show impulsive outliers? Is it possible to adopt your training tricks to LCM? In addition, since iCT is not open-source, I recommend the authors show more comparisons of your model and the original CM. For example, what about L2 loss and LPIPS loss in Table 2 (b)?

3. The ablation study to the hyper-parameters is not sufficient. Please provide more discussion on why you choose such a schedule in equation (11).

4. The presentation of this paper is somehow repetitive and complicated. For example, Lines 164-175 are quite similar to the abstract and introduction part. And sections 4.4 and 4.6 are all descriptions with pure texts. Introducing more equations or figures may help to improve the presentation of this paper.

**Questions:**

1. What is the correlation between the major contributions?
2. Could you please provide more quantitative comparisons of your models with LCM and original CM?
3. Why do you choose such a schedule in equation (11)?

**Details Of Ethics Concerns:**

No ethics review is needed.

---

> ### Author Response · Authors · 2024-11-23
>
> We thank the reviewer for their detailed and thorough review of our paper.
>
> Firstly, regarding Weakness 2, we would like to clarify that our implementation follows the original CM implementation. We only added new techniques, such as weighting, noise-level sampling, and the pseudo-Huber loss function from iCT paper. We carefully reviewed our implementation against the iCT to ensure its correctness.
>
> Following your suggestion, we trained the original CM model from scratch on the latent CelebA-HQ dataset for 1,400 epochs. Despite our efforts, the model failed to converge for both L2 and ELPIPS loss functions. (We used ELPIPS [1] since the latent space has 4 channels, which is incompatible with the LPIPS model.) We also tried setting the EMA rate to 0, as suggested by iCT, but the CM model still did not converge. It is worth noting that the original CM requires to initialized consistency model with pretrained diffusion model, making consistency training is more like consistency finetuning.
>
> To further assess the performance of L2 and ELPIPS losses, we trained our model with these two loss functions and reported the results in Table 2c of the revised submission. The L2 loss, being vulnerable to outliers, only achieved an FID of 50, whereas the ELPIPS loss performed significantly better with an FID of 11.49. However, ELPIPS has limitations: it requires training a model for each autoencoder, making it less broadly applicable. Additionally, ELPIPS relies on a neural network to compute the loss, resulting in longer training times of approximately 48 hours on an A100 GPU, compared to 42 hours with the Cauchy loss.
>
> [1]: Distilling Diffusion Models into Conditional GANs
>
>
>
> **Q1 & W1**: What is the correlation between the major contributions?
>
> **Response**: Firstly, we conducted a detailed statistical analysis of latent consistency training and found that the latent space contains numerous impulsive outliers. Additionally, the temporal difference (TD) objective of consistency also exhibits impulsive outliers. While impulsive outliers are also present in TD loss within the pixel space—prompting iCT to propose the robust pseudo-Huber loss. However, the impulsive outliers of TD are even more severe in the latent space. To address this, we proposed the Cauchy loss as a more robust learning objective. To further suppress outliers, we introduced an adaptive $c$, which scales $c$ down proportionally to the variance of the TD term. Furthermore, we investigated the role of normalization layers, as these layers capture statistical information and can be sensitive to outliers during the training process. This led us to propose non-scaling layer normalization (NsLN). Together, the three main contributions—Cauchy loss, adaptive $c$, and NsLN—are closely interrelated and specifically designed to address the impulsive outlier problem.
>
> In addition, during the training of the consistency model, we observed that the consistency loss at early timesteps was very small, resulting in poor $x_0$-prediction performance at these timesteps and degradation in overall performance. To address this issue, we proposed using diffusion loss at early timesteps to improve $x_0$-prediction, as diffusion loss effectively learns prediction at early timesteps. By incorporating diffusion loss at early timesteps, we enhanced the stability of the training process.
>
> Finally, we proposed minibatch optimal transport (OT) to reduce variance during consistency model training. This method, previously used as a variance reduction technique for diffusion models, also boosts the performance of the consistency model.
>
> In summary, to stabilize consistency training, we introduced Cauchy loss, adaptive $c$, and NsLN to address impulsive outliers. Minibatch OT was used for variance reduction during training, while diffusion loss at early timesteps improved $x_0$-prediction.
>
> Similar to our work, iCT [2] proposed a series of training techniques for pixel consistency models, bridging the performance gap between diffusion and consistency models. EDM [3], on the other hand, introduced improved training techniques for diffusion models, such as preconditioning, weighting, and noise-level sampling. These training techniques not only enhanced diffusion model performance but also proved highly beneficial for subsequent diffusion training advancements. In our paper, we focus on stable training or latent consistency model and bridge to performance gap between latent diffusion and consistency. Our proposed technique are well related to each other and focus on target, which is training stability improvement. With our proposed technique, the performance of latent consistency model is significantly improved, nearly reach latent diffusion model. Therefore, we think our method could be helpful for subsequent latent consistency training works.
>
> [2]: Improved Techniques for Training Consistency Models
>
> [3]: Elucidating the Design Space of Diffusion-Based Generative Models

---

> ### Author Response · Authors · 2024-11-23
> **Following response**
>
> **Q2**: Could you please provide more quantitative comparisons of your models with LCM and original CM?
>
> **Response**: We applied our techniques to LCM by setting the EMA rate to $0$, utilizing Cauchy/Huber loss, incorporating diffusion loss at early timesteps, and employing OT coupling. We retained the original normalization layer and set $c=1e-5$. We report the FID and CLIP score on COCO-30K in below table.
>
> **FID table**
>
> | Model | NFE=1 | NFE=2 | NFE=4 |
> | --- | --- | --- | --- |
> | LCM-L2* | 35.36 | 13.31 | 11.10 |
> | Our+Huber | 24.17 | 12.89 | 11.21 |
> | Our+Cauchy | 23.32 | 12.72 | 10.96 |
>
> **CLIP score table**
>
> | Model | NFE=1 | NFE=2 | NFE=4 |
> | --- | --- | --- | --- |
> | LCM-L2* | 24.14 | 27.83 | 28.69 |
> | Our+Huber | 27.82 | 29.85 | 30.11 |
> | Our+Cauchy | 28.15 | 30.54 | 30.90 |
>
> (*): number from LCM paper
>
> With the EMA rate set to 0, the model converges faster, requiring only 20 hours to train on a single A100 GPU. For latent consistency distillation, the model is initialized with a well-trained diffusion model, and $x_{t_i}$ is obtained by denoising diffusion from $x_{t_{i+1}}$. Therefore, their training target $f_{\theta^-}(x_{t_i}, t_i)$ is more stable, leading to a TD loss with fewer impulsive outliers. This makes it easier to distill consistency from a diffusion model.
>
> We think that robust loss functions like Cauchy and Huber may not significantly improve the quality of latent consistency distillation, since the training process of consistency distillation is stable. However, other techniques, such as OT coupling and incorporating diffusion loss at early timesteps, could enhance the performance of consistency distillation.
>
> Our results, shown in the tables above, indicate that for 1-NFE, our FIDs are significantly better than the original LCM and slightly better for 2-NFE. However, 1-NFE sampling remains very blurry, even with our improved training techniques. Furthermore, our CLIP score is also better than the orignal LCM.
>
> **Q3**: Why do you choose such a schedule in equation (11)?
>
> **Response**: We recorded consistency training objective, the TD loss, at difference discretization stages and compute its statistical variance. Higher discretization indicates smaller gap between adjacency timesteps hence the loss is smaller. We supposed there was connection between losses’ variance and the threshold $c$ in robustness loss, hence we scaled down $c$ proportional to variance, starting from $c = c_0$. We then fitted the sequence
>
> We recorded the temporal difference term $f_\theta(x_{t_{i+1}}, t_{i+1}) - f_{\theta^-}(x_{t_{i}}, t_i)$ during training latent consistency model and compute its variance for each number of discretization step. We are aware that the variance reduce when the number of discretization step increases. We think any value of TD that is too large compared to variance suppose to be the outlier. Therefore, we reduce the $c$ proportional to the variance of TD to better control the robustness of model. At the beginning number of discretization step, we choose $c=c_0$. For next number of discretization step, we scale down the $c$ proportional to the variance. After getting list of $c$, we fit an equation which take number of discretization step as input and output the corresponding $c$.
>
> **W4**: The presentation of this paper is somehow repetitive and complicated. For example, Lines 164-175 are quite similar to the abstract and introduction part. And sections 4.4 and 4.6 are all descriptions with pure texts. Introducing more equations or figures may help to improve the presentation of this paper.
>
> **Response**: Thanks for pointing out our duplicate in paper. We already remove the duplicate section and carefully read through paper and rewrite the complex part in more simple manner. Following your suggestion, we also add figure to visualize the OT technique in the revised paper (see fig. 3). For the section 4.6, we provide the detail formula for both LayerNorm and Non-scaling LayerNorm for enhancing the readability.
>
> **W3**: Lack ablation study for hyperparameter
>
> **Response**: We provide more fixed $c$ ($0.01 \times c_0$ and $0.001 \times c_0$) scheduler in figure 4 and observe that our adaptive $c$ scheduler still performs better than all fixed $c$ schedulers.
>
> **I hope my response addresses your concerns. If there are any additional questions or issues, please feel free to let us know. We would be happy to provide further clarification or engage in further discussion.**

---

> ### Author Response · Authors · 2024-11-25
>
> Thank you for your time and effort in the review process and for your thoughtful comments, which have been invaluable in helping us polish our work. With only two days remaining in the rebuttal period, we wanted to check if our additional experiments, positive results, and responses have fully addressed your concerns and if you might consider revisiting your score.

---

> > ### Comment · Reviewer_n8Mw · 2024-11-26
> >
> > Thank you very much for your detailed rebuttal. After reading the rebuttal and the revised PDF, I think my concerns have been well addressed. I tend to increase my score and vote for accepting this paper.

---

> > > ### Author Response · Authors · 2024-11-26
> > > **Thank you for your time and consideration**
> > >
> > > Thank you for considering our rebuttal and raising the score. We appreciate your time and effort. Your feedback greatly helps us improve and refine our work.

---

### Official Review · Reviewer_7pHT · 2024-11-03

**Soundness:** 3
**Presentation:** 3
**Contribution:** 3
**Rating:** 8
**Confidence:** 4

**Summary:**

The paper proposes a new strategy for consistency training of latent consistency models. It analyzes the reasons for the poor performance of improved consistency training in latent space, particularly addressing the issue of outliers by suggesting the use of Cauchy loss as a remedy. Additionally, by using x0 as the ground truth when t is small, the paper reduces the errors caused by model fitting. Compared to improved consistency training, the proposed method shows significant improvements in latent space.

**Strengths:**

1.	The writing of the paper is clear.
2.	The paper analyzes the reasons for the poor performance of improved consistency training in latent space.
3.	The results of the proposed method achieved a great improvement than improved consistency training in latent space.

**Weaknesses:**

1.	Equation (9) should be ||f(xt)-x0||^2.
2.	Please briefly explain how the constant in Equation (11) is determined.
3.	Batch size has a significant impact on generative models, especially in consistency training; please include the batch size in Table 1.
4.	Please include a comparison of the results from latent consistency distillation, including the resources used for training.

**Questions:**

Please see the Weakness.

---

> ### Author Response · Authors · 2024-11-23
>
> Thank you for taking the time to provide a detailed review and for recognizing the challenges and performance aspects of training models in the latent space.
>
> **W1** Equation (9) should be $||f_\theta(x_t, t)-x_0||^2$.
>
> **Response**: Thanks for pointing our typo, we have corrected it in revised submission.
>
> **W2**: Please briefly explain how the constant in Equation (11) is determined.
>
> **Response**: We recorded the temporal difference (TD) term, $ f_\theta(x_{t_{i+1}}, t_{i+1}) - f_{\theta^-}(x_{t_i}, t_i) $, during the training of the latent consistency model and computed its variance for each number of discretization steps. We observed that the variance decreases as the number of discretization steps increases. We hypothesize that any TD value significantly larger than the variance should be considered an outlier. To address this, we adjust $c$ proportionally to the variance of the TD term to enhance the model's robustness. Initially, for the smallest number of discretization steps, we set $c = c_0$. For subsequent discretization steps, we scale $c$ down proportionally to the variance. Once we obtain the list of $c$ values, we fit an equation that takes the number of discretization steps as input and outputs the corresponding $c$.
>
> **W3**: Batch size has a significant impact on generative models, especially in consistency training; please include the batch size in Table 1.
>
> **Response**: We have provided the total training batch size in the table 1. (the total training batch size = batch size per gpus $\times$ gpus). In our paper, due to limited resource, we use 1 A100 for the CelebA-HQ and FFHQ with total batch size is 128. For Lsun Church, we use 2 A100s with total batch size 256.
>
> **W4**: Please include a comparison of the results from latent consistency distillation, including the resources used for training.
>
> **Response**: We apply the latent consistency distillation for our trained LDM-8 CelebA-HQ diffusion model and obtain the below results. The training time of LDM-8 for CelebA-HQ is around 38.59 gpu hours and the distillation time is around 2 hours. In contrast, our training time is around 42 hours. We can see that the performance of distillation model is bounded by the pretrained diffusion model. Furthermore, the latent consistency model produces very blurry generated images with 1-NFE sampling and this behaviour is also witnessed in works [1, 2, 3].
>
> |Model | NFE | FID |
> | --- | --- | --- |
> | LDM-8 | 256|8.85|
> | LCM | 1 | 22.19 |
> | LCM | 2 | 13.27 |
> | Ours | 1 | 7.27 |
> | Ours | 2 | 6.93 |
>
> [1]: Latent Consistency Models: Synthesizing High-Resolution Images with Few-Step Inference
>
> [2]: Hyper-SD: Trajectory Segmented Consistency Model for Effective Image Synthesis
>
> [3]: Trajectory Consistency Distillation: Improved Latent Consistency Distillation by Semi-Linear Consistency Function with Trajectory Mapping
>
> **I hope my response addresses your concerns. If there are any additional questions or issues, please feel free to let us know. We would be happy to provide further clarification or engage in further discussion.**

---

> ### Comment · Reviewer_7pHT · 2024-11-25
>
> After reading the rebuttal, my concerns have been addressed, so I raise the score.

---

> ### Author Response · Authors · 2024-11-25
> **Thank you for your time and contribution to review process**
>
> Thank you for considering our rebuttal and raising the score. We appreciate your time and effort.

---

### Official Review · Reviewer_2d6A · 2024-11-03

**Soundness:** 3
**Presentation:** 2
**Contribution:** 2
**Rating:** 6
**Confidence:** 3

**Summary:**

This paper presents a series of techniques for enhancing the training of consistent models in potential space, including Cauchy loss, early diffusion loss, and optimal transport coupling between noise and samples.

**Strengths:**

1. The motivation of this article are very sound and the introductory section does a good job of presenting the motivation and contribution of the article. Given the popularity of potential spatial diffusion modeling, this research may have important implications for the practical application of accelerated diffusion.

2. The authors found significant differences between latent space training and pixel space training. The latter is usually normalized and the former may have impulse noise. This factor may seem simple, but it has been long neglected.

**Weaknesses:**

1. The link to TD training in DQN in section 4.1 seems somewhat redundant. There is no evidence that DQN has a similar problem with impulse noise. And there is no solution proposed by the authors that is not derived from it.

2. The authors mention in the introduction that the aim is to address the potential proliferation of large-scale applications such as text-to-image or video generation. However, instead of using a text-to-image model like LCM, the authors ended up experimenting on some simple pattern datasets, which is a minor shortcoming of the experimental evaluation.

**Questions:**

In my experience, some losses that are robust to outliers usually slow down the convergence of the model, due to the fact that these losses are equivalent to reduced gradients in large error regions, which may lead to slower convergence in early training than standard losses. Have the authors observed this phenomenon in their experiments?

---

> ### Author Response · Authors · 2024-11-22
>
> Thank you for your thorough review and for recognizing the contributions and potential impact of our work. We truly appreciate your insightful feedback and support.
>
> **W1**: The link to TD training in DQN in section 4.1 seems somewhat redundant. There is no evidence that DQN has a similar problem with impulse noise. And there is no solution proposed by the authors that is not derived from it.
>
> **Response**: In this paper, we discuss about TD training in DQN as DQN also exhibits instability issues. Since the consistency model is also based on TD training and experiences similar instability, this motivated us to examine the TD loss as a potential source of the instability. Our analysis revealed impulsive outliers in TD values in both the pixel and latent spaces, with the outliers being more extreme in the latent space. Based on these observations, we propose robust training techniques, such as Cauchy loss and Non-scaling LayerNorm, to address the instability. While I acknowledge that there is no direct evidence of impulsive noise in DQN, I already revised Section 4.1 to clarify this point and avoid any confusion for readers.
>
> **W2**: The authors mention in the introduction that the aim is to address the potential proliferation of large-scale applications such as text-to-image or video generation. However, instead of using a text-to-image model like LCM, the authors ended up experimenting on some simple pattern datasets, which is a minor shortcoming of the experimental evaluation.
>
> **Response**: Thank you for your thorough comment. The motivation for investigating the performance of latent consistency training indeed stems from the needs of large-scale applications. Despite advancements like iCT, latent consistency training has demonstrated poor performance. Our primary goal was to analyze the root causes of this underperformance and propose effective solutions.
>
> We observed that the issue arises due to statistical discrepancies between the latent and pixel spaces and proposed techniques specifically designed to address this. While LCM focuses on consistency distillation by initializing models with diffusion models and using diffusion to sample $x_{t_i}$ from $x_{t_{i+1}}$, thereby avoiding instability issues, our work centers on training consistency models from scratch without leveraging any prior knowledge from diffusion models.
>
> As a small lab with limited resources, training large-scale models like text-to-image generation from scratch remains a significant challenge for us. Furthermore, we are the first to identify the failure cause of latent consistency training and propose effective solutions to address them. We hope our work serves as a foundational step toward developing large-scale text-to-image and text-to-video consistency models.
>
> **Q1**: In my experience, some losses that are robust to outliers usually slow down the convergence of the model, due to the fact that these losses are equivalent to reduced gradients in large error regions, which may lead to slower convergence in early training than standard losses. Have the authors observed this phenomenon in their experiments?
>
> **Response**: Thank you for your insightful observation. We have also observed the same behavior in our experiments. Please refer to the middle plot of Figure 3, where we visualize and compare FID scores during training progress between our proposed adaptive $c$ and several constant $c$ values. A smaller $c$ indicates a stronger robustness factor in the training objective (i.e., the Cauchy loss).
>
> Focusing on the two lines, the purple line ($c = 3.0 \times c_0$) and the green line ($c = 0.1 \times c_0$), we can see that using $c = 3.0 \times c_0$ achieves better FID scores in the early stages of training (e.g., at $36 \times 10^3$ iterations). However, this is eventually surpassed by $c = 0.1 \times c_0$, demonstrating the trade-off between early convergence speed and robustness to outliers over training process.
>
> **I hope my response addresses your concerns. If there are any additional questions or issues, please feel free to let us know. We would be happy to provide further clarification or engage in further discussion.**

---

> ### Author Response · Authors · 2024-11-25
>
> Thank you for your time and effort in the review process and for your thoughtful comments, which have been invaluable in helping us polish our work. With only two days remaining in the rebuttal period, we wanted to check if our additional experiments, positive results, and responses have fully addressed your concerns and if you might consider revisiting your score.

---

> > ### Comment · Reviewer_2d6A · 2024-11-26
> >
> > Thanks for addressing my concerns. I tend to keep my current score, which is already positive.

---

> > > ### Author Response · Authors · 2024-11-26
> > > **Thank you for your time and consideration**
> > >
> > > Thank you for considering our rebuttal. We appreciate your time and effort.

---

### Official Review · Reviewer_fHyc · 2024-11-03

**Soundness:** 2
**Presentation:** 3
**Contribution:** 2
**Rating:** 6
**Confidence:** 3

**Summary:**

The authors analyzed statistical differences between pixel space and potential space, and found that potential data often contained anomalous values of altitude pulses, which significantly reduced iCT performance in potential space. To solve this problem, they proposed the Cauchy loss, which effectively mitigated the effects of anomalies. Additionally, they introduce diffusion losses at early time steps and employ optimal transport coupling to further improve performance.

**Strengths:**

1. The analysis and motivation of this manuscript is sound, and to the best of my knowledge, the effect of the potential spatial anomalies they first revealed on consistent model training.

2. Each of the proposed techniques was well ablated.

3. From the visualizations provided by the authors (Figures 4,5), several of the proposed techniques do significantly improve the visuals of data such as Celeba-HQ.

**Weaknesses:**

1. As an empirical paper, the authors seem to have compared only with iCTs that reproduce in hidden spaces. In fact, there have been many improvements on the consistent model of lifting hidden spaces, such as [1, 2], with which authors should compare or discuss.

[1] Hyper SD: Trajectory Segmented Consistency Model for Effective Image Synthesis

[2] Trajectory consistency disruption

2. The authors' experiments are limited to some simple modal datasets such as the FFHQ, CELEBA-HQ datasets. Empirical evidence without multi-modal datasets weakens persuasion.

**Questions:**

1. The pseudo-huber loss also seems to be able to be characterized similarly to the Cauchy loss by adjusting c to adjust the degree of suppression of outliers. I'd like to know what the authors think.

2. ﻿ The batch for consistency model training is usually large. What is the increase in total time when using POT to calculate optimal transport for noise data coupling?

---

> ### Author Response · Authors · 2024-11-22
>
> Thank you for your detailed review and for posing interesting and thought-provoking questions.
>
> **W1**: As an empirical paper, the authors seem to have compared only with iCTs that reproduce in hidden spaces. In fact, there have been many improvements on the consistent model of lifting hidden spaces, such as [1, 2], with which authors should compare or discuss.
>
> **Response**:
>
> [1] Hyper-SD: Trajectory Segmented Consistency Model for Effective Image Synthesis
>
> [2] Trajectory Consistency Distillation: Improved Latent Consistency Distillation by Semi-Linear Consistency Function with Trajectory Mapping
>
> [3] Consistency Trajectory Models: Learning Probability Flow ODE Trajectory of Diffusion
>
> [4] Multistep Consistency Models
>
> The first two papers, [1] and [2], primarily focus on distillation from pretrained diffusion models—a scenario where the model already benefits from a strong starting point rather than random initialization, and these methods utilize pretrained diffusion models to sample $x_{t_{i}}$ from $x_{t_{i+1}}$, thereby avoiding training instability caused by impulsive noise.
>
> CTM [3] proposed an improved version of consistency models, achieving better performance in both training and distillation in the pixel space. TCD [2] extended CTM [3] to latent space distillation. Hyper-SD [1] segmented the PF-ODE into multiple components, similar to MCM [4], and applied TCD [2] to each segment. Hyper-SD [1] then progressively merged these segments into a final one, leveraging human feedback learning and score distillation to enhance one-step generation.
>
> While our approach is orthogonal to these works, it has the potential for integration with these methods. For example, our techniques could be applied to CTM [3], as their experiments are limited to the pixel space. Incorporating our method into these frameworks represents an exciting direction for future research. We already included this discussion in the updated version of the submission.
>
> **W2**: The authors' experiments are limited to some simple modal datasets such as the FFHQ, CELEBA-HQ datasets. Empirical evidence without multi-modal datasets weakens persuasion.
>
> **Response**: We appreciate your suggestion. Our work primarily aims to analyze the differences between the latent and pixel spaces for consistency training. Our latent consistency training has already demonstrated strong performance on several datasets, including FFHQ, CelebA-HQ, and LSUN-Church. Previous works, such as CM and iCT, have shown effectiveness on both single-modal datasets (e.g., LSUN-Bedroom and LSUN-Cat) and multi-modal datasets (e.g., ImageNet-1000). Therefore, we think our method could be scaled up to multi-modal datasets like ImageNet-1000. However, as a small lab, we currently lack the resources to conduct experiments on ImageNet-1000, especially given the time constraints of the rebuttal period.
>
> **Q1**: The pseudo-huber loss also seems to be able to be characterized similarly to the Cauchy loss by adjusting c to adjust the degree of suppression of outliers. I'd like to know what the authors think.
>
> **Response**:
>
> When residuals are within the range defined by $c$ (i.e., around zero), both loss functions behave similarly to the L2 loss. However, their behaviors differ when residuals exceed this threshold:
>
> - **Pseudo-Huber Loss:** For residuals outside the range of $c$, the pseudo-Huber loss transitions smoothly and behaves similarly to the L1 loss. This means it linearly penalizes larger residuals, providing a balance between L1 and L2 losses.
> - **Cauchy Loss:** In contrast, the Cauchy loss becomes a logarithmic function of the L2 loss when residuals exceed $c$. This logarithmic nature offers even greater robustness to extreme outliers compared to the pseudo-Huber loss.
>
> We might reduce $c$ in pseudo-Huber to very small value to obtain the effective training as Cauchy. However, for very extreme outliers like latent consistency training, the Cauchy loss should be suitable candidate loss function.
>
> **Q2**:  The batch for consistency model training is usually large. What is the increase in total time when using POT to calculate optimal transport for noise data coupling?
>
> **Response**: We measured the total training time across various batch sizes to determine the additional time introduced by noise data coupling with POT. The experiments were conducted on the same device, an A100 40GB GPU, using the CelebA-HQ 256 setting. We record the total training hours for 1400 epochs with different batch size as below:
>
> | Batch size | w/o POT | w/ POT |
> | --- | --- | --- |
> | 64 | 59.11 | 59.50 (+ 0.66%) |
> | 128 | 41.61 | 42.00 (+ 0.93%) |
> | 192 | 35.78 | 36.17 (+ 1.09%) |
>
> **I hope my response addresses your concerns. If there are any additional questions or issues, please feel free to let us know. We would be happy to provide further clarification or engage in further discussion.**

---

> ### Author Response · Authors · 2024-11-25
>
> Thank you for your time and effort in the review process and for your thoughtful comments, which have been invaluable in helping us polish our work. With only two days remaining in the rebuttal period, we wanted to check if our additional experiments, positive results, and responses have fully addressed your concerns and if you might consider revisiting your score.

---

> ### Comment · Reviewer_fHyc · 2024-11-27
>
> I thank the authors for addressing my concerns about lossy design and computational overhead and recognize that the authors' insights can be useful to the consistency model community. I understand the authors' lack of computational resources, but this is a point where the manuscript could be enhanced. Overall, I am inclined to keep the current positive score.

---

> > ### Author Response · Authors · 2024-11-27
> > **Thank you for your time and consideration**
> >
> > Thank you for considering our rebuttal. We appreciate your time and effort.

---

### Meta-Review · Area_Chair_Pfze · 2024-12-19

**Metareview:**

This paper introduces a novel strategy for training latent consistency models. The experiments demonstrate that the proposed strategy effectively enables high-quality sampling in just one or two steps, significantly reducing the performance gap between latent consistency models and diffusion models. The paper is well-written, well-organized, and makes a valuable empirical contribution to the community. A noted limitation is that, due to constraints in computational resources, the method was not evaluated on large-scale applications. However, the rebuttal systematically addressed most of the concerns raised by the reviewers in their initial assessments. This led to a consensus among all four reviewers, who agreed to accept the paper based on its innovation and empirical contributions. The AC concurs with the reviewers and recommends accepting the paper.

**Additional Comments On Reviewer Discussion:**

Reviewer fHyc highlighted the need for comparisons with additional baselines and noted the lack of testing on multimodal datasets. In the rebuttal, the authors explained that the suggested methods are orthogonal to their current submission and acknowledged that due to limited computational resources, they were unable to scale up to multimodal datasets. Reviewer fHyc also raised questions about the pseudo-Huber loss and the computation time for different batch sizes. The authors provided explanations and experimental results to address these queries. After the rebuttal, Reviewer fHyc maintained a rating of 6, which is marginally above the acceptance threshold.

Reviewer 2d6A expressed similar concerns regarding the lack of testing on more complex datasets and raised questions about the model's convergence. The authors addressed these major concerns, reiterating that computational constraints prevented them from conducting multimodal experiments. Reviewer 2d6A also maintained a rating of 6, which is marginally above the acceptance threshold.

The authors provided clarifications and additional experiments to address all questions raised by Reviewer 7pHT, who subsequently increased their rating and recommended accepting the paper.

Reviewer n8Mw initially raised concerns about the paper's contributions, insufficient comparisons, and the lack of an ablation study. The authors responded with detailed explanations and additional results to address these issues. After the rebuttal, Reviewer n8Mw increased their score and voted to accept the paper.

Overall, a consensus was reached among all four reviewers, culminating in the decision to accept the paper.

---

### Decision · Program_Chairs · 2025-01-22

Accept (Poster)